# The Knowledge Connector decision support system for multiomics-based precision oncology

Daniel Hübschmann [1,2,3,4,5,13] ✉, Simon Kreutzfeldt [3,6,13], Benjamin Roth [7,13], Katrin Glocker[7,8,13], Janine Schoop[7], Lena Oeser[7], Steffen Hausmann[7], Christian Koch[7], Sebastian Uhrig [1], Jennifer Hüllein[1], Barbara Hutter [1], Martina Fröhlich[1], Christoph E. Heilig [6], Maria-Veronica Teleanu[6], Daniel B. Lipka [3,6], Irina A. Kerle [9,10,11], Annika Baude-Müller[6], Katja Beck[6], Christoph Heining[9,10,11], Hanno Glimm [9,10,11], Frank Ückert[12], Alexander Knurr[7,14], Stefan Fröhling [3,5,6,14] ✉ & Peter Horak [3,6,14] ✉

Precision cancer medicine aims to improve patient outcomes by providing individually tailored recommendations for clinical management based on the evaluation of biological disease profiles in multidisciplinary molecular tumor boards (MTBs). The quality of MTB decisions depends on the comprehensive, reliable, and reproducible interpretation of increasingly complex molecular data. We developed and implemented, as part of a multicenter precision oncology program, the Knowledge Connector (KC), a decision support system that integrates individual patients' molecular and clinical data with world knowledge to generate and document MTB recommendations. The KC supports data curation, database integration, and discussion based on multiomics data and provides an interface for creating a cross-institutional knowledge base. Furthermore, it extracts relevant biomarker-drug associations and increases the efficacy of data interpretation in a clinically relevant manner by reducing reliance on external sources and optimizing inter-curator concordance. Our results demonstrate that the KC is a versatile tool that supports medical decision-making in MTBs, thus enabling the scalability of precision cancer medicine.

Technologies that enable comprehensive molecular analysis of individual patients' tumors at high-throughput have led to the emergence of precision oncology. This approach is based on identifying, annotating, and evaluating a broad spectrum of molecular biomarkers ranging from single-gene mutations to complex profiles, such as mutational or gene expression signatures, and composite scores combining several of these measures. Such biomarkers aim to capture the predictive, prognostic, and diagnostic value of molecular data to categorize patients into different treatment and outcome groups.

Well-established examples from clinical practice include *BRAF* V600E mutations, which predict response to inhibition of BRAF−MEK signaling in melanoma[1], or *NPM1*, *FLT3*, and *CEBPA* mutations, which provide prognostic information in acute myeloid leukemia[2].

Currently, DNA-based sequencing of up to several hundred genes is the most widely used diagnostic tool in precision oncology. However, there is increasing evidence that whole-genome sequencing (WGS), whole-exome sequencing (WES), RNA sequencing (RNA-seq), and probably other omics layers have additional clinical value[3–7].

A full list of affiliations appears at the end of the paper. ✉e-mail: d.huebschmann@dkfz-heidelberg.de; stefan.froehling@nct-heidelberg.de; peter.horak@nct-heidelberg.de

Therefore, complex multilayered data must be considered in clinical decision-making, which represents a new challenge in oncology[8]. A critical step to overcome this challenge has been the establishment of dedicated precision oncology workflows that feed into molecular tumor boards (MTBs), where experts from various disciplines, including oncology, medical genetics, bioinformatics, molecular biology, pathology, and others, discuss individual cases. However, expert opinions alone cannot achieve the necessary standardization, reproducibility, and scalability.

The first step in a precision oncology workflow is the molecular characterization of tumor tissue using high-throughput methods, which are becoming increasingly numerous and comprehensive. For example, as part of the MASTER (Molecularly Aided Stratification for Tumor Eradication Research) program of the German Cancer Research Center (DKFZ), the National Center for Tumor Diseases (NCT), and the German Cancer Consortium (DKTK), advanced cancers are routinely subjected to WGS or WES, RNA-seq, and DNA methylation profiling[6,9]. Raw data are processed bioinformatically and forwarded to biomedical curators for functional assessment, followed by the interpretation of specific alterations regarding their oncogenicity and the selection of clinically actionable biomarkers, which are discussed in an interdisciplinary MTB.

The next step is data curation, which aims to obtain a humanly comprehensible set of biomarkers, followed by the preparation, execution, and documentation of MTBs. These workflow components are not standardized, leading to relevant differences between institutions in the practical use of biomarker information and inconsistencies in the reporting of molecular data[10–12]. Consequently, the main bottleneck of MTB workflows is curating and interpreting molecular alterations. This is a complex process in which experts manually search knowledge bases for information on specific variants and corresponding therapeutic options. However, the range of sources and the large amount of information make systematic and time-efficient processing difficult, and the content of knowledge bases is heterogeneous[13]. Furthermore, currently available knowledge bases are organized by genes or biomarkers[14,15]. While such a structure covers many applications in precision oncology, there is a growing number of complex biomarkers and alterations at the supragenic level, e.g., polyploidy, large copy number aberrations, and mutational signatures. Another challenge is that evidence in the medical literature usually applies to a specific clinical setting. In precision oncology, an individual case may have some overlap but not an exact match with published evidence.

The ambition to provide clinical care tailored to the unique characteristics of each individual cancer has entailed several new challenges. In addition to considering precision oncology in training programs and the definition of guidelines and standard operating procedures, these mainly include the development of decision support systems, which have become one of the most pressing goals of clinical informatics. These software solutions make molecular tumor profiles accessible and interpretable for data curators and medical professionals by linking them with relevant, accurate, and up-to-date content from knowledge bases and play an increasing role in expediting, standardizing, and ensuring the quality of reporting within and between institutions[16].

Here, we present the development and implementation of the Knowledge Connector (KC), a web application to support MTB workflows, within the DKFZ/NCT/DKTK MASTER precision oncology network, which uses multiomics for clinical decision-making (https://demo.kc.dkfz.de; code available at https://github.com/CTO-SUDO/de.dkfz.esr.knowledgeconnector (KC Frontend) and https://github.com/CTO-SUDO/de.dkfz.esr.worker.knowledge (KC Backend)). We describe how the KC framework facilitates linking individual patients' molecular profiles with world knowledge, including clinical reasoning based on complex biomarkers and evidence chains. We further introduce the concept of blocks of clinical knowledge (BoCKs) to map evidence to individual biomarkers and explain that the KC also serves as an interface for creating a cross-institutional knowledge base, the BoCKbase.

## Results

### Development of the KC as a decision support system for MTBs

The identification and interpretation of molecular biomarkers is a multistep process[17]. To support several of these steps, particularly biomarker annotation, oncogenicity classification, clinical interpretation, and clinical decision-making (Fig. 1a and Supplementary Fig. 1), we developed the KC. The KC (https://demo.kc.dkfz.de; code available at https://github.com/CTO-SUDO/de.dkfz.esr.knowledgeconnector (KC Frontend) and https://github.com/CTO-SUDO/de.dkfz.esr.worker.knowledge (KC Backend)) is a web-based, European Union General Data Protection Regulation-compliant system that allows protected access to pseudonymized clinical and molecular data and supports the fast, accurate, and complete execution of MTB workflows using comprehensive molecular data, e.g., from WGS and RNA-seq. The home screen displays meta-information about the tumor sample and patient and condensed clinical data. In a curation and interpretation process guided by a navigation bar at the top of the page, users can select different views ranging from analyzing integrated biomarkers and biomarker scores, inspecting individual biomarkers, evaluating pharmacogenomic information, curating and suggesting treatment recommendations, to presenting the data in the MTB (Fig. 1b). The KC thus combines essential clinical and complex molecular data in an intuitive and comprehensive format.

Specifically, the main view of the KC consists of three sections (Fig. 1b):

1. To the left of the screen, a vertical bar displays meta-information about the tumor sample and clinical data, such as patient identifier, date of birth, diagnosis, or relevant previous treatments.
2. At the top of the screen, a navigation bar enables essential steps in MTB preparation:
   a. *Integrated biomarkers*: Information and underlying evidence on complex and composite biomarkers that exceed the single-gene level, e.g., tumor mutational burden (TMB), mutational signatures, or quantitative measures of genomic instability
   b. *Molecular biomarkers*: Information and underlying evidence on clinically actionable biomarkers at the single-gene level
   c. *Pharmacogenomic information*: Recommendations of the Clinical Pharmacogenetics Implementation Consortium for selected pharmacogenes
   d. *Recommendations*: Grouping and enrichment of information selected in steps a. and b., assignment of treatments to biomarkers, and assessment of molecular evidence levels (mEL)[18]
   e. *Presentation*: Compilation of the selected biomarkers and recommended treatments for the MTB, final assignment of mEL, and ranking of recommendations
3. The largest part of the screen is a frame for displaying the molecular data, enriched with world knowledge selected via the navigation bar. This adjustable frame may contain tables, graphics, text, editable text fields, pull-down menus, etc.

For the annotation of relevant molecular findings, comprehensive resources, including, but not limited to, Ensembl[19], OncoKB[20], CIViC[15,21], Reactome[22], and JAX-CKB[23], or individual evidence items, e.g., from PubMed, are used. To optimize the integration of comprehensive molecular and clinical data from individual patients with world knowledge (Fig. 1c), we have introduced the concept of BoCKs. These aggregated statements cite diagnostic, prognostic, predictive, or clinical trial evidence for a particular biomarker consistent with a patient's tumor entity (Fig. 1d and Supplementary Fig. 2a). Clinical

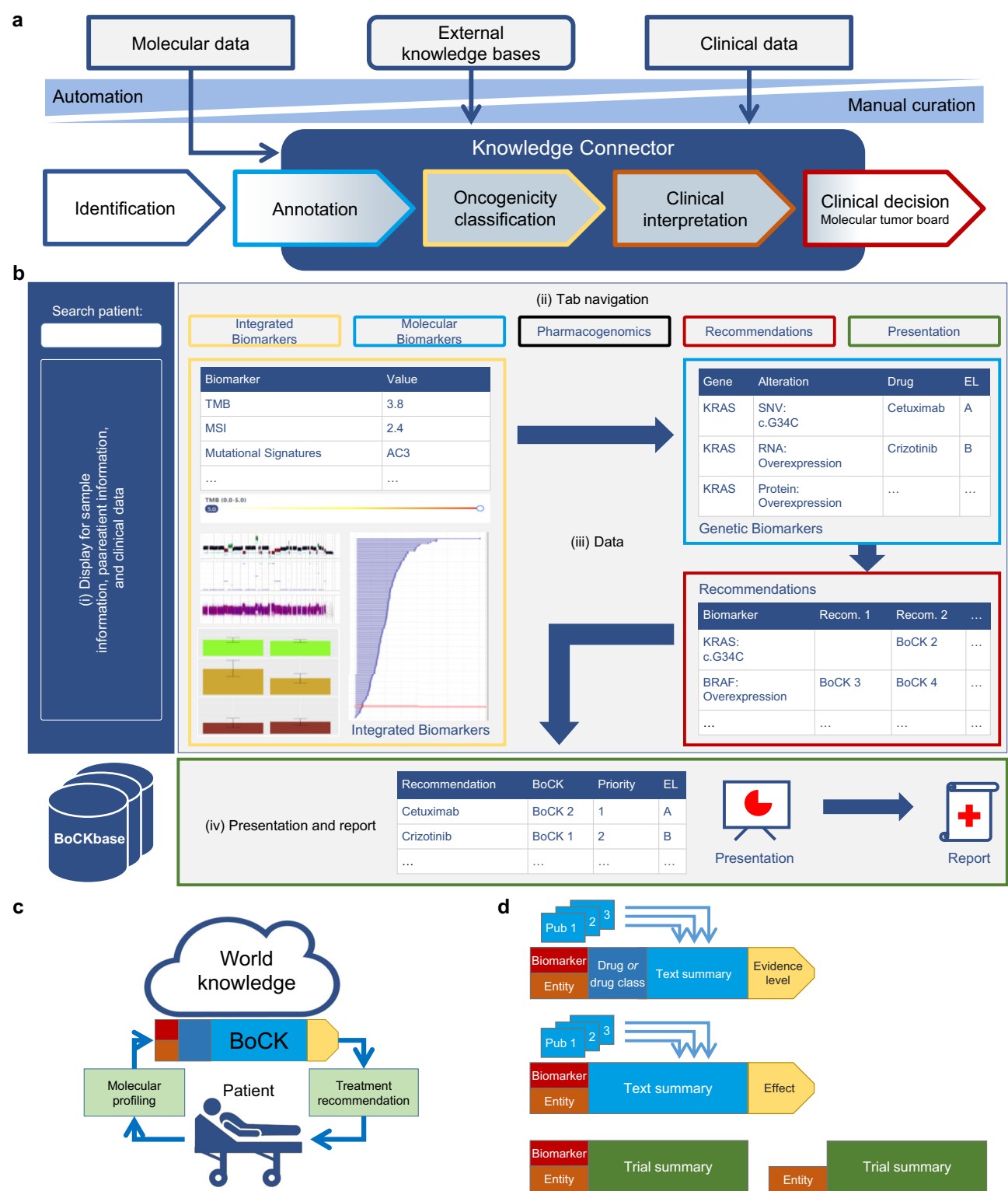

Fig. 1 | **Scope and functionality of the KC. a** Precision oncology workflow steps supported by the KC. **b** Schematic overview of the KC, including (i) meta-information on the tumor sample and the patient and clinical data, (ii) the navigation bar, (iii) content selected via the navigation bar, and (iv) the presentation step consisting of a series of slides and concluding with an editable MTB recommendation and prioritization table and a downloadable report. **c** Link between molecular and clinical data from individual patients with world knowledge via BoCKs. **d** Examples of BoCKs. Top, BoCK combining a biomarker (e.g., a specific gene mutation), an entity, a drug or drug class, and published evidence, resulting in a predictive statement with a specific evidence level (EL). Middle, BoCK combining a biomarker, an entity, and published evidence, resulting in oncogenicity classification. Bottom, BoCK combining a biomarker, an entity, and published evidence, identifying eligibility for a specific clinical trial. AC COSMIC (Catalogue Of Somatic Mutations In Cancer) mutational signatures version 2, MSI microsatellite instability, Pub publication, Recom. recommendation, SNV single-nucleotide variant, TMB tumor mutational burden.

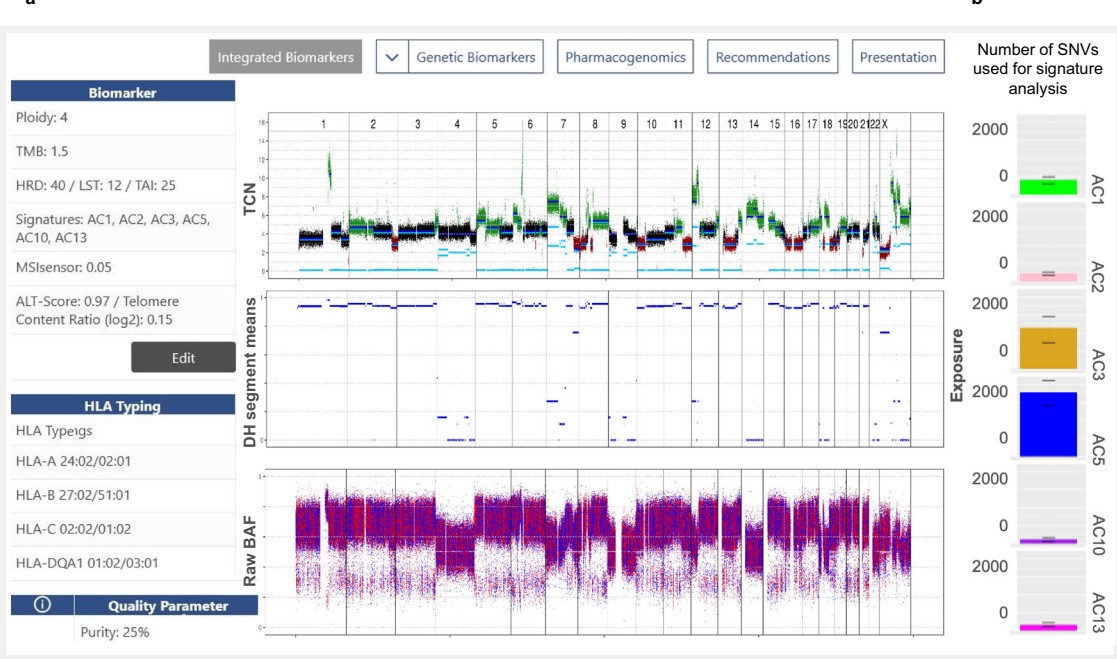

**Fig. 2 | KC display of HRD and mutational signatures. a** Left, complex bio-markers, displayed via selection of the "Integrated Biomarkers" tab in the navigation bar, with numerical values for HRD, LSTs, and TAI to quantify genomic instability. Display frame on the right, visualization of total copy number (TCN), decrease of heterozygosity (DH), and B-allele frequency (BAF). **b** Visualization of exposures to mutational signatures as alternative content of the display frame; barplots display the exposures, whiskers indicate 95% confidence intervals as computed by profile likelihood. AC mutational signature, ALT alternative lengthening of telomeres, *HLA* human leukocyte antigen, HRD homologous recombination deficiency score, LST number of large-scale state transitions, TAI quantification of telomeric allelic imbalances, TMB tumor mutational burden.

decisions often consider indirect matches, e.g., when a particular molecular alteration is oncogenic (Fig. 1d and Supplementary Fig. 2b) or predictive in another tumor entity, or by extending the search for a specific drug to other compounds with similar pharmacodynamic effects. BoCKs can accommodate this type of reasoning and also include information on biomarker-stratified clinical trials (Fig. 1d and Supplementary Fig. 2c). All curated BoCKs are stored in a database, the BoCKbase, and are reusable for future curation, generating a growing resource for molecularly informed clinical decision-making at the individual-patient level.

## Use of the KC for MTB preparation
To illustrate the KC's functionality and unique features in assessing comprehensive, multidimensional tumor profiles, we selected cases from the MASTER program[6,9] with biomarkers not usually detected by standard technologies, i.e., DNA-based panel sequencing.

### Homologous recombination deficiency
Homologous recombination deficiency (HRD) can leave various imprints on the genomes of cancer cells, including specific mutational signatures, such as single-base substitution signature 3 (refs. 24–26), and quantifiable manifestations of genomic instability[27–29]. Such imprints, which predict sensitivity to poly(ADP-ribose) polymerase (PARP) inhibitors or platinum-based chemotherapy, are often detected in the absence of "classical" loss-of-function mutations in genes associated with homologous recombination, such as *BRCA1/2*. The KC provides an integrated and intuitive display of the two genomic imprints of HRD, accessible via the "Integrated Biomarkers" tab. Genomic instability is indicated by three parameters (Fig. 2a): the loss of heterozygosity (LOH)-HRD score[27], the number of large-scale state transitions (LSTs)[29], and the amount of telomeric allelic imbalance (TAI)[28]. Exposure to mutational signatures computed with YAPSA[25] is

displayed as stacked barplots. Figure 2b shows an HRD phenotype in a 43-year-old woman with undifferentiated thyroid carcinoma. This unexpected finding in this entity illustrates the functionality of the KC as a visualization and decision support tool and prompted screening for participation in a molecularly stratified clinical trial of combination treatment containing the PARP inhibitor olaparib (ClinicalTrials.gov: NCT03127215).

### Gene fusions
The KC also allows the visualization and reporting of RNA-seq-based gene fusion calls of the software Arriba[30]. The "Genetic Biomarkers" view of the KC integrates all alteration types affecting a gene of interest, e.g., *BRAF* in a 59-year-old male with parotid gland carcinoma harboring a previously unknown *AGK::BRAF* fusion that resulted from a genomic inversion reported as a structural variant (SV; Fig. 3a). Transcriptomic analysis confirmed the fusion of *AGK* as the N-terminal and *BRAF* as the C-terminal partner. The KC provides a graphical representation of several features of the fusion derived from Arriba, i.e., the location of the breakpoints relative to the coding sequence, the exons involved, the orientation (sense or antisense) of the fusion product (Fig. 3b), the chromosomal position (Fig. 3c), and the preserved functional domains of the resulting protein (Fig. 3d). The *AGK::BRAF* fusion involved *BRAF* from exon 8, including the kinase domain from exon 11, and retained the open reading frame. Therefore, the fusion was considered to activate the MAPK pathway. This evaluation was supported by gene-level information from internal and external knowledge bases, summarized in BoCKs with different mEL. Preclinical work (mEL 3) showed an effect of the MEK1/2 inhibitor trametinib in *AGK::BRAF*-positive melanoma models[31]. In addition, *AGK::BRAF* fusions have been detected in cancer patients, and response to MEK inhibition has been observed in individual cases (mEL 2C)[32–34]. For example, a patient with *AGK::BRAF*-driven melanoma showed

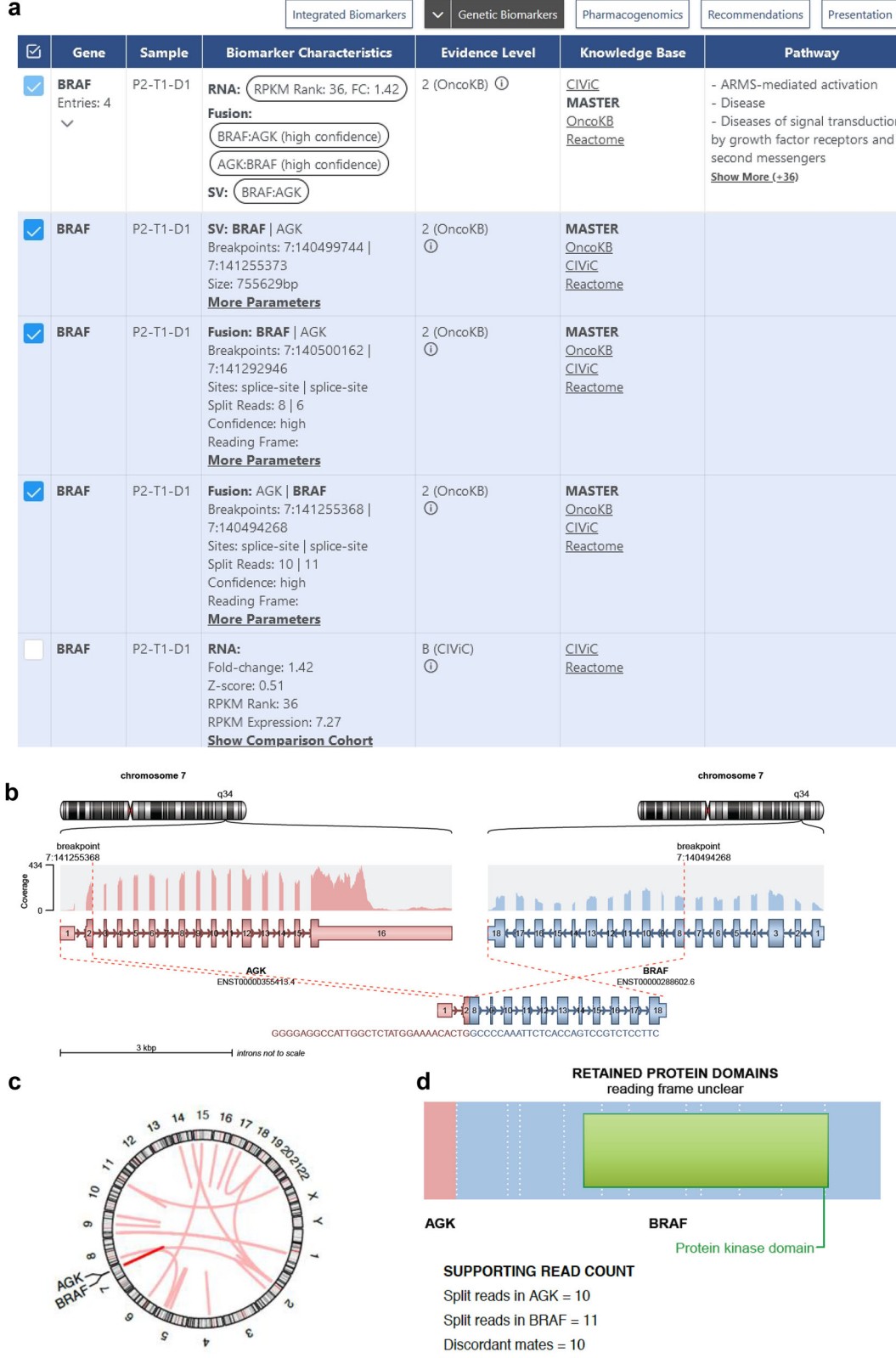

**Fig. 3 | KC display of an AGK::BRAF fusion. a** Screenshot of the "Genetic Biomarkers" panel illustrating the congruent detection of a genomic SV and an overexpressed fusion transcript. **b** Visualization of the AGK::BRAF fusion event by the fusion detection tool Arriba. **c** Chromosomal position of the AGK::BRAF fusion. **d** Domain structure of the resulting AGK::BRAF fusion protein. FC fold change, KC Knowledge Connector, kbp kilobase pair, RPKM reads per kilobase of transcript per million mapped reads, SV structural variant.

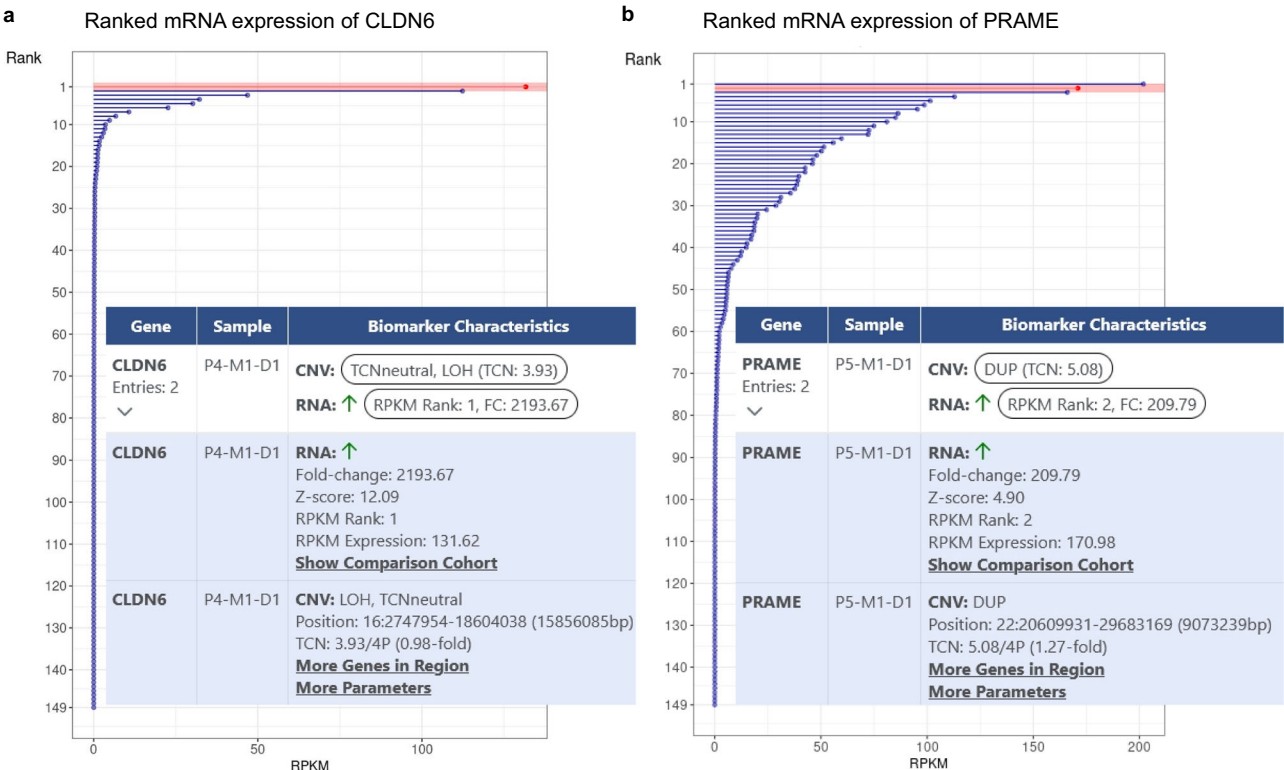

**Fig. 4 | KC display of expression biomarkers. a** CLDN6 mRNA expression. The expression rank of the patient described is indicated in red. Inset, screenshot of the "Genetic Biomarkers" panel illustrating CLDN6 overexpression. **b** PRAME mRNA expression. The expression rank of the patient described is indicated in red. Inset, screenshot of the "Genetic Biomarkers" panel illustrating PRAME overexpression and a DNA copy number gain in the corresponding genomic region. CNV copy number variant, DUP duplication, FC fold change, RPKM reads per kilobase of transcript per million mapped reads, TCNneutral copy number neutral LOH, 4P inferred ploidy of 4.

resistance to vemurafenib but sensitivity to sorafenib in vitro and subsequently a durable response to sorafenib in vivo[32], and a patient with *AGK::BRAF*-positive myoepithelial carcinoma responded to cobimetinib[34]. Given these BoCKs, one MTB recommendation was sorafenib and/or a MEK inhibitor. Another example of fusion visualization in the KC is a 38-year-old female with melanoma and a *TRIM33::BRAF* fusion between two amplified genomic regions (TCN of 8 in an inferred diploid genome; Supplementary Fig. 3a). The fusion product included exons 1–11 of *TRIM33* and *BRAF* from exon 9 (Supplementary Fig. 3b). This fusion was one of several detected (Supplementary Fig. 3c), and its clinical significance was readily identified using the KC, as the predicted chimeric protein included the kinase domain and retained the BRAF open reading frame (Supplementary Fig. 3d). Based on these findings and supported by several BoCKs, pan-RAF and/or MEK inhibition were recommended by the MTB.

### Aberrant gene expression

The KC also allows the integration and exploratory analysis of data generated with emerging technologies such as transcriptomic, epigenomic, and proteomic profiling. For example, RNA-seq is becoming increasingly important for clinical decision-making as, beyond gene fusion detection, it allows the quantification of mRNA expression of structurally intact genes and their isoforms, whose protein products are targeted by a growing number of therapies. Figure 4a shows the case of a 42-year-old patient with a mediastinal non-seminomatous germ cell tumor characterized by overexpression of *CLDN6* (2,193-fold, rank 1 of the reference subcohort). Based on this finding, treatment with CLDN6-directed chimeric antigen receptor T cells was recommended (mEL 1 C)[35,36] as part of a clinical trial (NCT04503278). Another example is a 36-year-old patient with advanced Ewing sarcoma and a 210-fold (rank 2) increase in mRNA expression of the *PRAME* cancer

testis antigen (Fig. 4b). PRAME is a potential target for the development of cellular immunotherapies in sarcoma (mEL 4)[37], while in medulloblastoma, preclinical studies have shown the efficacy of adoptive immunotherapy with PRAME-specific T cells (mEL 3)[38]. Analogous to the example of CLDN6, at the time of the MTB, a matching clinical trial (NCT03686124) was recruiting, whose eligibility criteria included *HLA A*02:01* positivity, which was confirmed by in silico *HLA* typing from DNA sequencing data and validation using orthogonal methods. The KC also displays such information to the MTB, enabling rapid evaluation of a patient's potential eligibility for clinical trials.

### Curation of clinical biomarker associations

Once the relevant molecular alterations are identified, the curation process proceeds to the assignment of recommendations and evidence defined in BoCKs. In the corresponding KC view, all predictive, prognostic, and diagnostic biomarkers are displayed in a compact form, and the clinical consequences, including therapeutic intervention, genetic counseling, or pathologic reevaluation, can be determined. If one or several BoCKs show biomarker-drug associations, multiple drugs and/or drug classes[39] (Supplementary Fig. 4a) and clinical trial BoCKs (Supplementary Fig. 4b) can be assigned to the recommendation. The KC aggregates information from available databases and automatically matches it to the selected biomarker. However, since information obtained from different knowledge bases is heterogeneous and automatic matching is unreliable[13], subsequent human curation, correction, and confirmation are required. Figure 5 depicts the process of curating an evidence-based MTB recommendation. Based on the selection of relevant biomarkers, BoCKs are used to formulate one or more recommendations, which are assigned a cumulative, preliminary molecular evidence. In the second step, an analogous assignment is made for clinical trials, which can only be

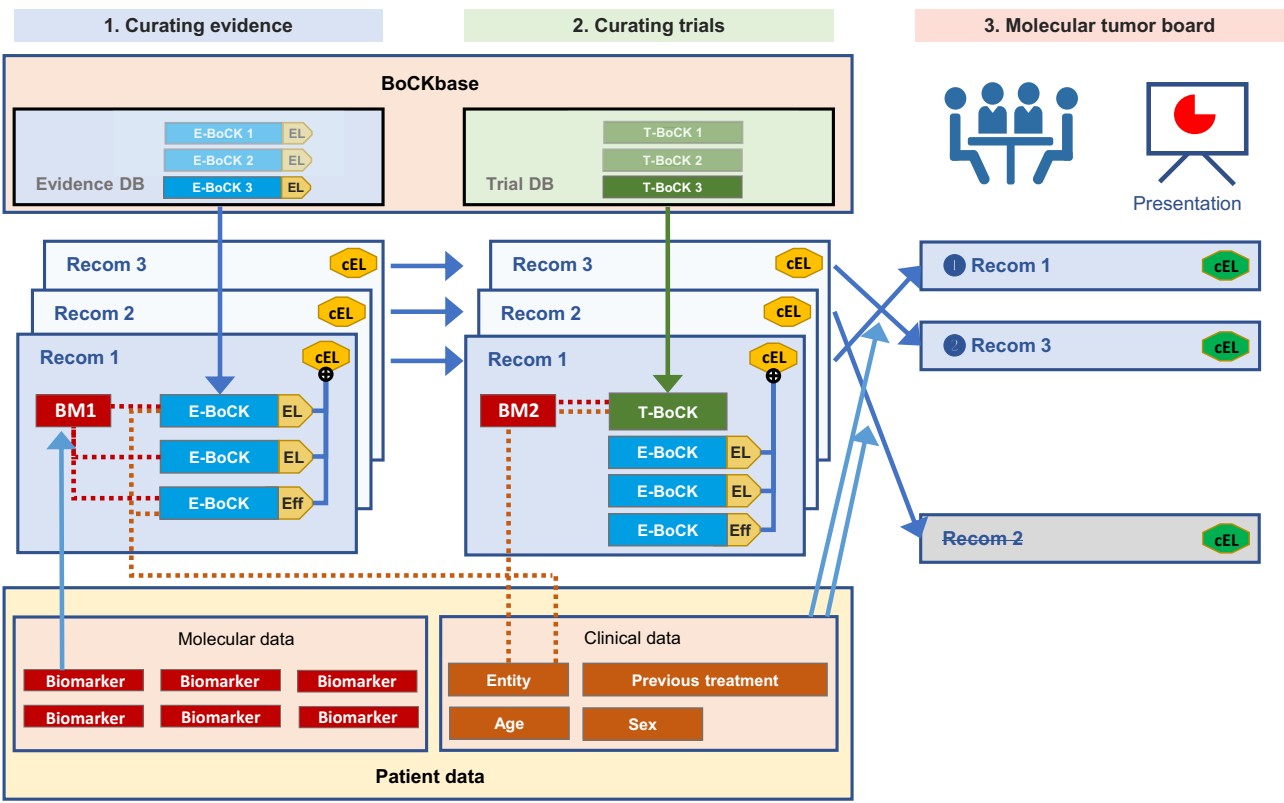

**Fig. 5 | Curation of evidence-based MTB recommendations.** Curating recommendations for individual patients in precision oncology involves three steps. For each recommendation, clinical and molecular data (bottom) are linked to BoCKs (top). The first step is to link to medical interventions, followed by clinical trials. The third step comprises the presentation and clinical decision-making, including prioritization of recommendations, in the MTB. BM biomarker, cEL clinical evidence level, DB database, E-BoCK evidence block of clinical knowledge, Eff effect, EL evidence level, Recom recommendation, T-BoCK clinical trial block of clinical knowledge.

partially automated and must consider patient eligibility and trial availability. In the final step, the curated recommendations for clinical management and trial enrollment are presented to and discussed by the MTB. As a visualization and decision support system, the KC interacts with a continuously growing, manually curated database for BoCKs, the BoCKbase. The BoCKbase can be made accessible to multiple users and institutions, making it a user-maintained, quality-controlled resource that evolves and leads to a community consensus through the use of the KC in MTB preparation.

### Real-world application of the KC

The KC has been in productive use at NCT Heidelberg since February 2022 and has since been implemented at several other comprehensive cancer centers in Germany. To evaluate its usability and utility in clinical settings, we conducted a structured online survey among active KC users ($n = 21$), receiving 15 fully completed responses. The survey was analyzed using established criteria for clinical decision support system evaluation[40]. Respondents were primarily medical oncologists ($n = 9$), molecular biologists ($n = 2$), and human geneticists ($n = 2$) (Supplementary Fig. 5a), most of whom had several years of experience in biomedicine and multiomic data curation (Supplementary Fig. 5b–d). KC usage experience ranged from less than 1 year (novice; $n = 8$) to more than one year (advanced; $n = 7$), with a median of 50 (range, 0–600) self-reported curated cases (Supplementary Fig. 5e, f). Most users emphasized the need for decision support systems to aid clinical interpretation of complex multiomics datasets. The KC was positively evaluated for its visualization capabilities and comprehensive data integration (Supplementary Fig. 5g, h), although some limitations in the user interface were noted (Supplementary Fig. 5i). Users reported improved data curation efficiency, resulting in

workflow enhancements related to MTB presentation, discussion, and documentation (Supplementary Fig. 5j–l). Adoption of the KC was generally rapid, with users quickly becoming familiar with the system and indicating a high likelihood of recommending it to colleagues (Supplementary Fig. 5m, n). No statistically significant differences (two-sided Wilcoxon rank sum test) were identified between novice and advanced users.

A detailed analysis of the first 268 curated KC cases examined the distributions of recommendations, actionable biomarkers, BoCKs, and clinical trials assigned per patient (Supplementary Fig. 6). At least one recommendation for clinical intervention was made in 243 of 268 patients (90.7%; median, 3; range, 1–7). A suitable molecularly stratified clinical trial was identified in 151 of 268 cases (56.3%). The high yield of biomarkers per patient (median, 5; range, 1–16) reflects the breadth of information contained in comprehensive molecular profiles, and the assignment of up to 20 BoCKs per patient (median, 7) demonstrates the utility of this condensed format for biomarker curation and information transfer to the MTB. Figure 6a illustrates the evolution of key MTB metrics during the KC adoption period, including case volumes, the number of therapeutic recommendations, as well as biomarkers and clinical trial recommendations per case. No immediate or clinically relevant changes were observed in these parameters over time. A decline in clinical trial recommendations was observed, likely due to reduced local trial availability.

The number of newly generated BoCKs stored in the BoCKbase is steadily increasing (Fig. 6b). As expected, it was initially proportional to the number of processed cases but decoupled after approximately four months, demonstrating the reuse of previously stored BoCKs. This decoupling reflects the potential of BoCKs developed and maintained as part of biomedical curation in a clinical setting to create a

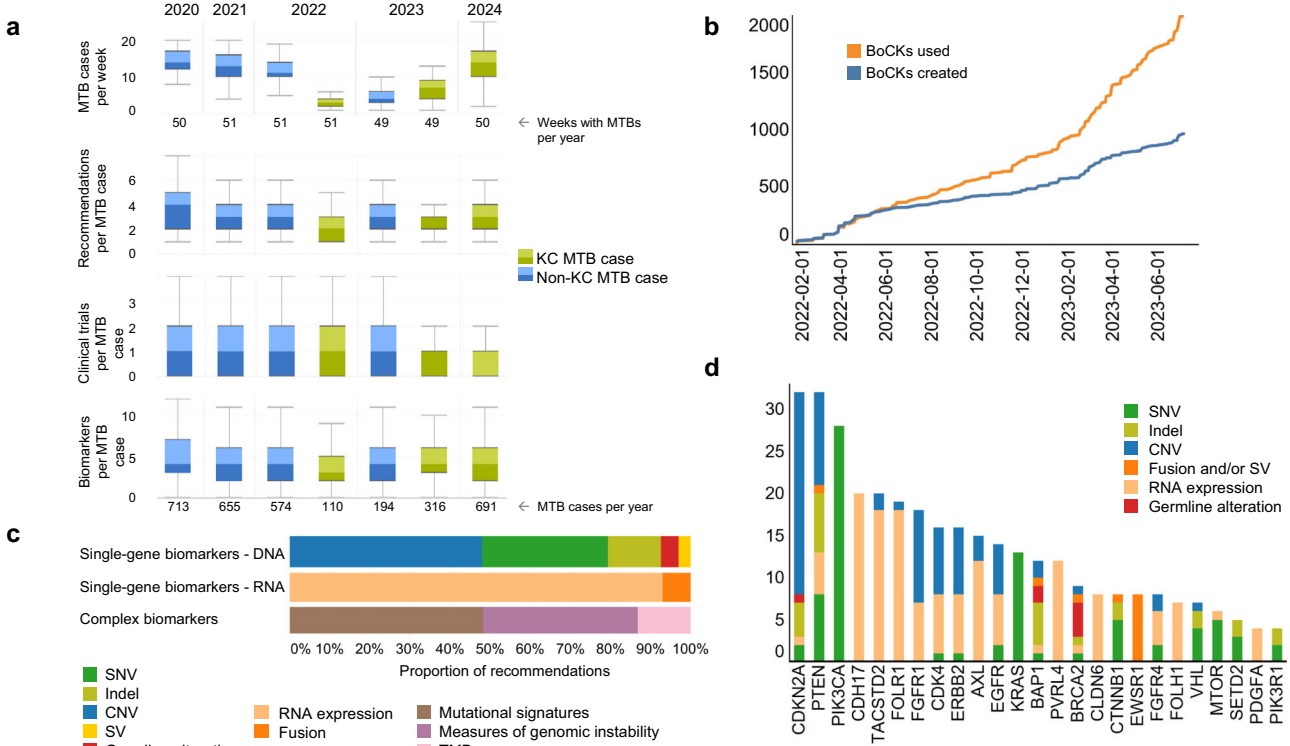

**Fig. 6 | Overview of MTB cases curated using the KC. a** Development of key MTB characteristics during the KC adoption period. The boxplots show the distribution of the number of MTB patients per week, treatment recommendations per case, clinical trial recommendations per case, and the number of biomarkers used per case between 2020 and 2024. In each boxplot, the center line indicates the median, and the boxes represent the interquartile range (IQR). The change in color hue shows the lower and upper IQR, and the whiskers extend to the furthest data point within 1.5 times the IQR of the box. Numbers of data items are displayed below the respective boxplots. **b** Cumulative number of BoCKs created in the BoCKbase and used for clinical decision-making over time. **c** Proportion of single-gene and complex biomarkers. **d** The twenty most frequently used biomarker genes and their molecular alterations. Indel, small insertion/deletion. BoCK block of clinical knowledge, CNV (somatic) copy number variation, KC Knowledge Connector, MTB molecular tumor board, SNV single-nucleotide variant, TMB tumor mutational burden.

knowledge base that increases the efficiency and scalability of precision oncology workflows. The molecular alterations used for clinical recommendations include a broad spectrum of single-gene and complex biomarkers (Fig. 6c). Most DNA-based single-gene biomarkers are SNVs and somatic CNVs, of which those affecting *CDKN2A*, *PTEN*, and *PIK3CA* are most commonly used for MTB decisions (Fig. 6d). At the RNA level, aberrant expression of individual genes dominates (Fig. 6c), often prompting the recommendation of targeted therapies, e.g., against CDH17, FOLR1, or TACSTD2 (Fig. 6d). Complex biomarkers comprise mutational signatures, measures of genomic instability, and TMB estimates (Fig. 6c).

## Discussion

We here describe the KC, a data integration, curation, visualization, and decision support platform for precision oncology workflows. By providing a flexible and user-friendly interface, the KC reduces manual workload, thereby facilitating clinical decision-making in MTBs and increasing their scalability.

Specific tasks in the KC workflow require expertise from different domains and thus the involvement of multiple people in the multistep process of curating molecular data for decision-making in MTBs, which includes variant calling and the identification, annotation, functional classification, and clinical evaluation of biomarkers. While variant calling and the identification and annotation of biomarkers can be standardized and automated, functional and clinical evaluation remain manual tasks. Evidence for the relevance of a specific biomarker relies, with few exceptions, on unregulated external knowledge bases and

needs to be organized and presented to the MTB in a concise and patient-centric manner. Clinical guidelines and standard operating procedures for these tasks are being developed but are highly heterogeneous across institutions and countries[41]. The KC, a software tool for extracting actionable information from multilayered molecular cancer profiles, supports several steps of this process by providing annotation and structure for the functional and clinical assessment of biomarkers, considering the specifics of the diagnostic technologies applied. It aggregates information from multiple sources and provides biomedical curators with all necessary algorithmic functions for sorting, filtering, and the functional assessment of biomarkers using various views to optimally inform molecularly guided treatment decisions. In addition, bioinformatic quality controls for the approval of biomarkers are part of the KC.

Of note is that the KC supports recommendations for clinical management based on single-gene as well as complex and composite biomarkers. The underlying data structures are flexible, allowing for rapid integration and development of new, user-defined composite biomarkers, whose relevance increases as molecular testing becomes more comprehensive. The KC decision support system can already accommodate biomarkers based on a wide range of molecular information layers, i.e., the genome (including germline variants associated with hereditary cancer predisposition or pharmacogenomic risk), transcriptome, DNA methylome, and phosphoproteome, enabling multiomics integration at the patient level. Ongoing developments focus on user role development and incorporating data generated by additional tumor profiling methods, such as digital

pathology approaches and functional drug screens, or new bioinformatics tools.

Another essential feature of the KC is its database functionality, which is based on the concept of BoCKs. While the BoCKbase complements rather than replaces external knowledge bases, it enables the merging of evidence items from different sources based on diverse data models, thus facilitating the reuse and storage of BoCKs. Another strength of BoCKs is the generation of content for medical reports in the working language of the respective MTB, facilitating implementation in non-English-speaking countries. In addition, the BoCKbase can be implemented and maintained across institutions, thereby leveraging the collective expertise of precision oncology networks.

Several academic and commercial efforts aim to develop cross-institutional MTB platforms[16,42,43]. However, few studies have directly addressed their usability in clinical settings[44]. The KC can be compared with several both academic and commercial decision support tools, including widely used knowledge bases, with regard to core genetic data and knowledge base features (Supplementary Tables 1 and 2). To further evaluate its clinical utility, we conducted a structured user survey to gather feedback on the current use of the KC at our institutions. The KC has two distinct features that make it particularly suited to address the challenge of processing multilayered molecular profiles: (a) by focusing on the curation of biomarkers and their mapping to drugs without committing to specific bioinformatic functions, it is technology-agnostic and can be adapted to any portfolio of molecular profiling methods; (b) by balancing the level of automation to increase productivity and not forcing algorithmic decisions, the KC is open to ongoing developments and the implementation of standard operating procedures for biomarker annotation and functional classification. For precision oncology to reach its full potential, patients' molecular profiles must also be linked to information from electronic health records (EHRs) and previous MTB evaluations to create a sustainable resource of biomarker profiles, treatment recommendations, and, ideally, patient outcomes. Substantial limitations persist regarding standardization and generalization, both regionally and globally, including variations in EHR formats and content, application programming interface design, and, to a lesser extent, molecular data formats. Even though challenging, sharing such data collections across institutions will also help standardize MTB recommendations and enable less experienced physicians and their patients to benefit from the expert knowledge generated.

In summary, we have developed a point-of-care software solution that extracts clinically relevant biomarker-drug associations from comprehensive, multilayered molecular cancer profiles, increases the efficiency of data curation and interpretation, and links molecular and clinical data at the individual-patient level. Integrating the KC into precision oncology workflows will substantially increase their quality, standardization, and throughput, which is essential for the widespread adoption of molecularly informed cancer medicine in an increasing number of patients.

## Methods

Patients in the MASTER program provided written informed consent for banking of tumor and control tissue, molecular analysis, and the collection of clinical data under a protocol (S-206/2011) approved by the Ethics Committee of the Medical Faculty of Heidelberg University. The study was conducted in accordance with the Declaration of Helsinki.

### Patient-related data

Patients' medical histories are obtained from EHRs based on data points defined by medical oncologists, bioinformaticians, and medical informaticians, i.e., patient and biospecimen identifiers, date of birth, sex, diagnoses, topologies and morphologies of the primary tumor and metastases, therapies administered, and response and survival data. Molecular data from, e.g., WGS or WES and RNA-seq, are obtained as processed data files from the bioinformatics workflow. Raw sequencing data are analyzed by a set of pipelines operated by the One Touch Pipeline, an automated workflow and data management system[45]. Briefly, WGS and WES data are aligned with a workflow based on bwa[46], and variant calling is performed using mpileup for germline and somatic SNVs, platypus for germline and somatic indels, ACEseq[47] for somatic CNVs from WGS, cnvkit[48] (RRID:SCR_021917) for somatic CNVs from WES, and SOPHIA for germline and somatic SVs. RNA-seq data are aligned with STAR[49] (RRID:SCR_004463), transcripts are quantified using featureCounts (RRID:SCR_012919), and gene fusions are called by Arriba[30] (RRID:SCR_025854). All bioinformatics pipelines have been benchmarked and described previously[6,50].

### Communication between the KC and other software

The KC uses as input a patient's clinical and molecular data on the one hand and world knowledge on the other. Each interface is implemented as a REST API. Clinical and molecular data are transferred (Supplementary Fig. 1) via the KC Data Pool to the KC Database, which consists of a PosgreSQL database and a data model specified for the KC (Supplementary Fig. 7 and Supplementary Tables 3–26). The individual elements of the data model do not enforce a controlled vocabulary, providing a flexible interface to accommodate different ontologies and classifications. Clinical data from various EHR systems can be integrated through transformation into the existing data model of the KC. Additional bioinformatics pipelines can be integrated by adding the appropriate tables to the existing data model. Queries with patient-related clinical and molecular data can be sent to the KC Database via one of the REST APIs to retrieve the content required for clinical decision-making in the MTB. To link clinical and molecular data with world knowledge, the KC performs fully or semi-automated weekly downloads of five external knowledge bases (Ensembl (RRID:SCR_002344), OncoKB (RRID:SCR_014782), CIViC, Reactome (RRID:SCR_003485), JAX-CKB; Supplementary Table 27), and MTB-relevant data are transformed into a relational database model and stored in the shared BoCKbase. Before that, comparable information from the different knowledge bases is harmonized. For this purpose, data fields are combined, and uniform terminology is applied while ensuring that the original data record can always be traced back to the respective knowledge base. Like the KC Data Pool, the BoCKbase consists of a PosgreSQL database, and the harmonization and transfer of data are done via SQL.

### Enrichment of patient-related data

Patient-related data in the KC Data Pool are enriched overnight with data from the BoCKbase and transferred to a third PosgreSQL database, the KC Patient Database, for display in the KC frontend. Application interfaces (REST API) are implemented for communication and data exchange between the databases, which are also used by the KC frontend to retrieve the data to be displayed from the KC Patient Database and the BoCKbase (Supplementary Fig. 4). Various SQL methods, including views, materialized views, and insert and update statements, are used to store data in the KC Patient Database. Due to the large number of BoCKbase records, especially patient-related data, the enrichment process is very time-consuming.

A second way to enrich patient-related data with BoCKs is directly driven by the users while curating a case in the KC. Curators can create new BoCKs for molecular biomarkers identified in patient samples at the level of integrated biomarkers, genetic biomarkers, or recommendations. These entries are stored in the BoCKbase and labeled as KC-generated entries. After review by an independent expert curation team, the flag is removed, and the new entries become full BoCKs accessible to other KC users. By keeping patient data and BoCKs separate, the KC is also suitable in a multisite setting

where a shared BoCKbase is maintained and used (Supplementary Fig. 8).

### Generation and storage of therapy recommendations

The result of the KC-based curation process is a set of clinical recommendations presented to the MTB along with the underlying biomarkers and supporting BoCKs. These recommendations are stored in the KC Patient Database, not as individual SQL parameters but as a case-specific JSON file containing all information required to describe a specific recommendation.

### KC backend and frontend

The KC backend and frontend are programmed in Java and Jakarta Server Faces. The PrimeFaces framework is used for the frontend. The ranked mRNA expression is visualized in an R-based Shiny app.

### Public KC instance

The KC and the five use cases presented can be explored in a public instance, which can be accessed by opening an account at registration.public.kc.dkfz.de. The account will be active for four weeks, and the number of simultaneous users can be limited. The public instance has limitations compared to the one in productive use at DKFZ since February 2022. The BoCKbase contains no data from or links to commercial software or databases. All other external knowledge bases are regularly updated. Only selected, representative BoCKs related to the biomarkers described in the use cases are included. The public KC instance provides all functions essential for MTB preparation, e.g., for creating BoCKs or curating and documenting recommendations. To prevent the public instance from deviating from its original state due to external entries, the KC Database is reset to its original state every Sunday at 1 pm Central European Time.

## Data availability

The KC is a web-based application designed to support MTB workflows by integrating multiomics data for clinical decision-making. A publicly accessible demo instance is available at https://demo.kc.dkfz.de. To ensure compliance with data protection and patient privacy regulations, the demo version contains only mock patient data, which can be explored via the same link. As this work does not constitute a clinical trial, concepts such as randomization, control groups, blinding, or power analysis are not applicable.

## Code availability

The source code for the demo instance is available via GitHub in the following repositories: SQL scripts of the KC databases: https://github.com/CTO-SUDO/KC-Databases. Software: KC Frontend: https://github.com/CTO-SUDO/de.dkfz.esr.knowledgeconnector. KC Backend: https://github.com/CTO-SUDO/de.dkfz.esr.worker.knowledge. BoCKbase: https://github.com/CTO-SUDO/de.dkfz.esr.bockbase. MASTER WDB Worker: https://github.com/CTO-SUDO/de.dkfz.esr.worker.kc_wdb. Own libraries: CTO SUDO parent: https://github.com/CTO-SUDO/de.dkfz.esr.parent. CTO SUDO transmission library: https://github.com/CTO-SUDO/de.dkfz.esr.transmissionlibrary. CTO SUDO JSON library: https://github.com/CTO-SUDO/de.dkfz.esr.json_library. CTO SUDO KC DTOs: https://github.com/CTO-SUDO/de.dkfz.esr.kc_dto.

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

## Acknowledgements

The authors thank the DKFZ/NCT/DKTK MASTER team, the NCT Heidelberg Sample Processing Laboratory, the DKFZ Next Generation Sequencing Core Facility, and the DKFZ Omics IT and Data Management Core Facility for workflow development, technical and administrative support, and data curation and management within the DKFZ/NCT/DKTK MASTER program. The MASTER program is supported by the NCT Overarching Clinical Translational Trial Program, the NCT Heidelberg Molecular Precision Oncology Program, and DKTK. This work was supported by the NCT Molecular Precision Oncology Program and the DKTK Molecularly Targeted Therapy Program.

## Author contributions

S.F., P.H., A.K., and D.H. designed the study. D.H., S.K., J.H., B.H., M.V.T., and P.H. defined requirements and formulated feature requests. B.R., K.G., J.S., L.O., S.H., C.K., S.U., F.Ü., and A.K. developed software. S.K., J.H., B.H., M.F., C.E.H., M.V.T., D.B.L., I.A.K., A.B., C.B., C.H., H.G., S.F., and P.H. provided data. D.H., S.K., B.R., K.G., and P.H. analyzed the data. D.H., S.K., B.R., K.G., A.K., S.F., and P.H. drafted the manuscript. All authors critically revised and contributed to the manuscript and approved its final version.

## Funding

## Competing interests

D.H.: stock: Platomics; S.K.: consulting or advisory board membership: Roche; honoraria: Roche; D.B.L.: honoraria: Illumina, Infectopharm; C.H.: consulting or advisory board membership: Boehringer Ingelheim; honoraria: Roche, Novartis; research funding: Boehringer Ingelheim; S.F.: consulting or advisory board membership: Illumina; travel or accommodation expenses: Illumina; P.H.: consulting or advisory board membership: Platomics; honoraria: Platomics, Roche, Trillium. The remaining authors declare no competing interests.

## Additional information

[1]Computational Oncology Group, Molecular Precision Oncology Program, National Center for Tumor Diseases (NCT), NCT Heidelberg, a partnership between the German Cancer Research Center (DKFZ) and Heidelberg University Hospital, Heidelberg, Germany. [2]Pattern Recognition and Digital Medicine Group, Heidelberg Institute for Stem Cell Technology and Experimental Medicine (HI-STEM), Heidelberg, Germany. [3]German Cancer Consortium (DKTK), DKFZ, Core Center Heidelberg, Heidelberg, Germany. [4]Innovation and Service Unit for Bioinformatics and Precision Medicine, DKFZ, Heidelberg, Germany. [5]Institute of Human Genetics, Heidelberg University Hospital, Heidelberg, Germany. [6]Division of Translational Medical Oncology, DKFZ, and NCT Heidelberg, Heidelberg, Germany. [7]Secondary Use of Data in Oncology Group, Clinical Trial Office, DKFZ, Heidelberg, Germany. [8]Medical Genetics Center, Munich, Germany. [9]Department of Translational Medical Oncology, NCT, NCT/University Cancer Center (NCT/UCC) Dresden, a partnership between DKFZ, Faculty of Medicine and University Hospital Carl Gustav Carus, TUD Dresden University of Technology, and Helmholtz-Zentrum Dresden-Rossendorf (HZDR), Dresden, Germany. [10]Translational Medical Oncology, Faculty of Medicine and University Hospital Carl Gustav Carus, TUD Dresden University of Technology, Dresden, Germany. [11]DKTK, Partner Site Dresden, Dresden, Germany. [12]Institute for Applied Medical Informatics, Medical Center Hamburg-Eppendorf, Hamburg, Germany. [13]These authors contributed equally: Daniel Hübschmann, Simon Kreutzfeldt, Benjamin Roth, Katrin Glocker. [14]These authors jointly supervised this work: Alexander Knurr, Stefan Fröhling, Peter Horak. ✉e-mail: d.huebschmann@dkfz-heidelberg.de; stefan.froehling@nct-heidelberg.de; peter.horak@nct-heidelberg.de

