## [Transparent Peer Review file · Nature Communications]

Knowledge Connector: Decision support system for multiomics-based precision oncology

Corresponding Author: Dr Daniel Hübschmann

Version 0:

Reviewer comments:

Reviewer #1

(Remarks to the Author)
Summary

Hübschmann, Kreutzfeldt, Roth, and Glocker present a thoughtful and well-organized manuscript focused on decision support in precision oncology. They frame the impetus of informed decision-making in the context of molecular tumor boards (MTBs) and review the features of their tool, the Knowledge Connector (KC), in detail. Additionally, they provide compelling examples of specific clinical decisions aided by the KC throughout the manuscript, as well as how the KC has generally been integrated into the workflow at NCT Heidelberg.

Strengths

1. Well-written and clearly presented
2. Compelling clinical anecdotes, as well as generally summary of the first 268 cases of the MTB where the Knowledge Connector was used
3. The Knowledge Connector has broader utility outside the context of this study as a clinical decision-making tool

Weaknesses

1. It is unclear to me if there has been any meaningful benchmarking to, or comparison with, other similar tools. One example is the MoAlmanac (<https://www.nature.com/articles/s43018-021-00243-3>). The study would be strengthened by some comparison on this front
2. The methods state that “All other external knowledge bases are current as of April 1, 2024, and have not been regularly updated.” Are there plans to regularly incorporate updates in the future? If so, will any part of this be automated? With the rapid evolution of medical knowledge, a plan for regular updates should be made and articulated.

Summary and Recommendation

This is a high-quality manuscript describing a useful clinical decision support tool. My recommendation is for acceptance after the comments above are addressed.

(Remarks on code availability)

I have not reviewed the code in detail, but have viewed the repository which is well organized, documented, and accessible.

Reviewer #2

(Remarks to the Author)
Summary

This manuscript presents Knowledge Connector (KC), a decision support tool for precision oncology that provides a platform that integrates a patient's data, a dashboard that compares the patient's data to relevant clinical trials and studies, and a

system to produce recommendations for clinical management. The manuscript highlights the different views that KC has and provides information on its impact in clinical settings in Germany on 268 subjects in Heidelberg. Decision support systems like KC can provide the oncologist and clinical care team with access to the latest information and clinical trials information to facilitate care. The major evidence of its utility thus far is its use in clinical settings to assist clinicians and clinical care specialists in treating patients. This is a nicely written manuscript that highlights how this tool has great potential in improving medical care. To strengthen the manuscript and highlight the utility of KC to the research community, the authors should provide evidence of utility and added-value to convince clinical teams and institutions to adopt KC. The data in Figure 6 are a great summary of its use so far, but do not provide visibility into its added-value (i.e., a pre-/post- comparison of outcomes after implementation).

Major Revisions

1) A major challenge in addressing the utility of KC is there is not sufficient evidence to point out that the treatment that the 268 individuals received changed as a function of the care team using KC. The authors need to provide a way to quantify the impact and added-value of their tool to better convince potential users for adoption.

- For example, if the recommendations that the patients received would have been exactly the same, then KC provides no added value to the standard care.
- Alternatively, if the added-value is a reduction in time to recommendations, then this provides external clinicians and care teams with the opportunity to assess whether onboarding this tool will improve their work flows.
- Is there a tool to assess the impact of negative recommendations or uncertainty? Are there human-review safeguards that can be implemented against potential recommendations that might be harmful for a patient? Line 352-353 describes striking this balance - but the authors should expand more to make explicit the choices done to ensure patient safety.

Table 1B (Sutton et al)[1] provides a more comprehensive evaluation of decision support systems in a clinical setting. I would strongly recommend the authors review this and evaluate KC based on the relevant criteria as a way to provide the reader with a systematic cost-benefit analysis of their tool. This evaluation will provide the readers (and potential adopters) a robust assessment of the utility of KC.

[1] Sutton, Reed T., et al. "An overview of clinical decision support systems: benefits, risks, and strategies for success." *NPJ digital medicine* 3.1 (2020): 17.

KC (and similar tools) have the potential to help provide medical institutions within (and outside) Germany if it can become a unifying decision support tool across institutions as the authors nicely point out in the discussion. However, the second major challenge is that, as the authors explicitly point out in lines 322-323

"Clinical guidelines and standard operating procedures ... are highly heterogeneous across institutions and countries."

The authors should provide additional evidence supporting the notion that KC can be implemented across institutions, or if not possible at this point given current scope, then discuss limitations/challenges with generalizability across institutions and countries.

- Can the authors clarify if the real-world evidence presented in Figure 6 corresponds solely to NCT Heidelberg or is the data across different medical centers?
- Are there limitations with integrations with different EHRs systems and standardizing information across them that the authors have encountered, and can they provide approaches to address these challenges?
- Would these tools and the optimizing of treatments generalize to distinct patient populations across the world, and if not, what would it take for KC to provide precision oncology independent of the individual's country of origin? As supplementary table 21 shows, if the scores of gene expression are relative to 'reference' cohorts and reference cohorts vary differently from site/institution/country, then statistically they become difficult to compare against.

Minor Comments:

- Line 370: istance → instance

- Line 308-309: suggested revision change total n = XX to total n = XX /268 (yy%) to provide a percentage relative to total cases (268).

The demo provided in the link was not accessible for this reviewer at the time of review. Would you have an alternative way of accessing the demo?

(Remarks on code availability)

I tried multiple times to access the demo instance using the "Credentials for demo instance" and unfortunately kept getting the same error message (see below). If there are alternate ways to access, I would be interested in testing it out and exploring the web app!

The server is temporarily unable to service your request due to maintenance downtime or capacity problems. Please try again later.

Version 1:

Reviewer comments:

Reviewer #1

(Remarks to the Author)

The authors have addressed my concerns thoroughly.

(Remarks on code availability)

Reviewer #2

(Remarks to the Author)

I thank the authors for their effort to address the feedback provided. The manuscript and the evidence supporting KC is stronger, and this will only increase the utility and impact of their tool and study.

The first major point I highlighted was for the authors to provide evidence of the utility of KC. The authors have provided evidence in the form of a post-study survey (Supp Fig1), and Supplementary Tables 1-2 provide evidence for the readers to demonstrate where KC stands relative to existing tools. The survey data provides evidence of positive attitudes towards usage as a result of implementation (Fig 6A), indicating overall a range from moderate to very positive attitudes across the questions. However, an N=15 is relatively small compared to the 268 cases curated. Are the authors able to survey prior users as well? If this is possible, I strongly recommend doing so to strengthen this point.

The second major point I highlighted was for the authors to demonstrate generalizability & limitations. The authors have clarified that the 268 patients that were presented were pooled from 17 cancer centers. The authors also indicate that "the system has also been successfully implemented at the University Medical Center Hamburg-Eppendorf – an institution outside our core networks" providing evidence and potential for broader adoption. Regarding standardizations regionally and globally, the authors highlight how their software ecosystem is able to handle the different EHR ecosystems as well. I recognize that addressing all potential challenges is clearly beyond the scope of the manuscript. To this end, I suggest the authors broaden the discussion section explaining standardizations and generalization limitations both regionally and globally, and also suggest that the disclaimer in the KC web application be explicitly stated/included in the manuscript.

Minor Comments:

- Fig6A: I recommend making a line plot showing adoption of KC vs Non-KC (similar to Fig 6b) or alternatively adding confidence/variation bands to demonstrate the range of behavior.

- Supp Fig1 G-M: I recommend creating stacked bar charts and overlay the 'average' score per question to make it easier to read & obtain take away messages from survey respondents.

- Supp Fig 1F -- please change the axes to be 0-10, 10-100, 100-1000 so that the axes and the legend show the same information.

Summary and Recommendation

My recommendation is for acceptance after the comments above are addressed.

(Remarks on code availability)

I have successfully tested the demo instance that the authors provided.

Knowledge Connector: Decision support system for multiomics-based precision oncology

We thank the reviewers for their insightful and constructive comments, which have substantially improved this work. Enclosed is a revised manuscript that has been modified in accordance with their recommendations. Our specific responses to the comments are detailed individually below.

Reviewer #1 (Remarks to the Author)

Summary

Hübschmann, Kreutzfeldt, Roth, and Glocker present a thoughtful and well-organized manuscript focused on decision support in precision oncology. They frame the impetus of informed decision-making in the context of molecular tumor boards (MTBs) and review the features of their tool, the Knowledge Connector (KC), in detail. Additionally, they provide compelling examples of specific clinical decisions aided by the KC throughout the manuscript, as well as how the KC has generally been integrated into the workflow at NCT Heidelberg.

Strengths

1. Well-written and clearly presented
2. Compelling clinical anecdotes, as well as generally summary of the first 268 cases of the MTB where the Knowledge Connector was used
3. The Knowledge Connector has broader utility outside the context of this study as a clinical decision-making tool

Response: Thank you for this positive feedback.

Weaknesses

1. It is unclear to me if there has been any meaningful benchmarking to, or comparison with, other similar tools. One example is the MoAlmanac (www.nature.com/articles/s43018-021-00243-3). The study would be strengthened by some comparison on this front

Response: Thank you for this excellent suggestion. In the revised manuscript, we now discuss several published academic tools and commercial software. We have also included a comparative table positioning the Knowledge Connector alongside representative tools (**Supplementary Tables 1 and 2**), highlighting key similarities and differences. We believe these additions substantially strengthen the manuscript and appreciate the opportunity to improve it.

2. The methods state that “All other external knowledge bases are current as of April 1, 2024, and have not been regularly updated.” Are there plans to regularly incorporate updates in the future? If so, will any part of this be automated? With the rapid evolution of medical knowledge, a plan for regular updates should be made and articulated.

Response: Thank you for pointing out this inaccuracy in our previous statement. Since the original submission, we have implemented a comprehensive update protocol for the system, which is now described in the revised manuscript.

Summary and Recommendation

This is a high-quality manuscript describing a useful clinical decision support tool. My recommendation is for acceptance after the comments above are addressed.

Response: Thank you again for the positive and encouraging feedback.

Reviewer #1 (Remarks on code availability)

I have not reviewed the code in detail, but have viewed the repository which is well organized, documented, and accessible.

Reviewer #2 (Remarks to the Author)

Summary

This manuscript presents Knowledge Connector (KC), a decision support tool for precision oncology that provides a platform that integrates a patient’s data, a dashboard that compares the patient’s data to relevant clinical trials and studies, and a system to produce recommendations for clinical management. The manuscript highlights the different views that KC has and provides information on its impact in clinical settings in Germany on 268 subjects in Heidelberg. Decision support systems like KC can provide the oncologist and clinical care team with access to the latest information and clinical trials information to facilitate care. The major evidence of its utility thus far is its use in clinical settings to assist clinicians and clinical care specialists in treating patients. This is a nicely written manuscript that highlights how this tool has great potential in improving medical care. To strengthen the manuscript and highlight the utility of KC to the research community, the authors should provide evidence of utility and added-value to convince clinical teams and institutions to adopt KC. The data in Figure 6 are a great summary of its use so far, but do not provide visibility into its added-value (i.e., a pre-/post- comparison of outcomes after implementation).

Response: Thank you for this positive and constructive feedback.

Major Revisions

1. A major challenge in addressing the utility of KC is there is not sufficient evidence to point out that the treatment that the 268 individuals received changed as a function of the care team using KC. The authors need to provide a way to quantify the impact and added-value of their tool to better convince potential users for adoption.
 - For example, if the recommendations that the patients received would have been exactly the same, then KC provides no added value to the standard care.
 - Alternatively, if the added-value is a reduction in time to recommendations, then this provides external clinicians and care teams with the opportunity to assess whether onboarding this tool will improve their work flows.
 - Is there a tool to assess the impact of negative recommendations or uncertainty? Are there human-review safeguards that can be implemented against potential recommendations that might be harmful for a patient? Line 352-353 describes striking this balance - but the authors should expand more to make explicit the choices done to ensure patient safety.

Table 1B (Sutton et al)[1] provides a more comprehensive evaluation of decision support systems in a clinical setting. I would strongly recommend the authors review this and evaluate KC based on the relevant criteria as a way to provide the reader with a systematic cost-benefit analysis of their tool. This evaluation will provide the readers (and potential adopters) a robust assessment of the utility of KC.

[1] Sutton, Reed T., et al. "An overview of clinical decision support systems: benefits, risks, and strategies for success." *NPJ digital medicine* 3.1 (2020): 17.

Response: Thank you for raising this relevant point. Demonstrating added value for visualization and decision support systems is indeed challenging. Among the alternatives suggested, we analyzed molecular tumor board recommendations from 2020 to 2024 and report the differences in the updated manuscript, comparing key characteristics before and after Knowledge Connector implementation (updated **Fig. 6a**). In addition, we conducted a user experience survey among Knowledge Connector users, which are provided in the Supplementary Information (**Supplementary Fig. 1**). This survey addresses many of the benefits and risks of clinical decision support systems identified by Sutton et al. To further support this analysis, we have included a comparative overview of key genetic data and knowledge base features of several available academic and commercial clinical decision support systems in **Supplementary Tables 1 and 2**.

KC (and similar tools) have the potential to help provide medical institutions within (and outside) Germany if it can become a unifying decision support tool across institutions as the authors nicely point out in the discussion. However, the second major challenge is that, as the authors explicitly point out in lines 322-323

“Clinical guidelines and standard operating procedures ... are highly heterogeneous across institutions and countries.”

The authors should provide additional evidence supporting the notion that KC can be implemented across institutions, or if not possible at this point given current scope, then discuss limitations/challenges with generalizability across institutions and countries.

Can the authors clarify if the real-world evidence presented in Figure 6 corresponds solely to NCT Heidelberg or is the data across different medical centers?

Response: Thank you for the opportunity to clarify. The data now presented in **Figure 6** and **Extended Data Fig. 5** are from 268 patients enrolled in the MASTER program, a nationwide precision oncology platform that integrates 17 cancer centers through the German Cancer Research Center (DKFZ), the National Center for Tumor Diseases (NCT), and the German Cancer Consortium (DKTK). This framework provides centralized molecular diagnostics and operates a joint molecular tumor board. Thus, the evidence presented reflects input from multiple centers across Germany. Clinical and molecular data analysis using the Knowledge Connector was performed by a core team of translational oncologists at NCT Heidelberg and NCT Dresden. The ongoing implementation of the Knowledge Connector across additional centers – ultimately targeting network-wide adoption – aims to establish a federated system that enables all partner institutions to process data independently while maintaining standardized protocols and methodologies. We are actively expanding deployment to partner institutions in Augsburg (NCT) and Berlin (NCT and DKTK). Notably, the system has also been successfully implemented at the University Medical Center Hamburg-Eppendorf – an institution outside our core networks – demonstrating its adaptability and broad applicability. Based on our experience, technical barriers to wider adoption are minimal; the primary constraint is the availability of dedicated personnel to support local deployment.

Are there limitations with integrations with different EHRs systems and standardizing information across them that the authors have encountered, and can they provide approaches to address these challenges?

Response: The core function of the Knowledge Connector is to integrate a cancer patient’s molecular profile with global knowledge on biomarker-drug associations to generate optimal personalized treatment recommendations. In this respect, the system has no inherent analytical constraints, as it was specifically developed to incorporate diverse ‘omics layers into clinical decision-making. It can, therefore, process varying levels of input complexity, from comprehensive genomic and transcriptomic data to targeted gene panel sequencing results. Regarding clinical data and electronic health record (EHR) usage, the Knowledge Connector currently ingests this data type from a centralized, curated database. In our modular software ecosystem, the challenges posed by the diversity of EHR platforms are addressed by a separate module, independent of the Knowledge Connector. This modular design enables flexible and rapid adaptation to heterogeneous data formats, either manually or with the aid of large language models, an area of active development. However, these capabilities lie outside the scope of the Knowledge Connector and this manuscript.

Would these tools and the optimizing of treatments generalize to distinct patient populations across the world, and if not, what would it take for KC to provide precision oncology independent of the individual’s

country of origin? As supplementary table 21 shows, if the scores of gene expression are relative to 'reference' cohorts and reference cohorts vary differently from site/institution/country, then statistically they become difficult to compare against.

Response: Thank you for highlighting the important aspect of data standardization and comparability across populations. We believe that tools such as the Knowledge Connector can play an important role in harmonizing bioinformatic workflows and supporting the development of data-sharing infrastructures across institutions and countries. We fully agree that addressing these challenges will be essential for implementing the Knowledge Connector globally. At the same time, we see the tool itself as part of the solution. By providing a versatile platform for aggregating 'omics data in routine clinical practice, the Knowledge Connector contributes to the development of population-specific reference datasets and supports broader efforts toward future standardization.

Minor Comments

Line 370: istance → instance

Response: Thank you. We have corrected this error in the revised manuscript.

Line 308-309: suggested revision change total n = XX to total n = XX /268 (yy%) to provide a percentage relative to total cases (268).

Response: We agree and have modified the manuscript accordingly. The data are now presented in **Extended Data Fig. 5**.

The demo provided in the link was not accessible for this reviewer at the time of review. Would you have an alternative way of accessing the demo?

Response: We apologize for the service disruption. Our team has reset the server, and we are implementing a sustainable maintenance schedule to ensure consistent uptime moving forward. The demo instance is now available at <https://demo.kc.dkfz.de>. Login credentials: review (username), WmeWPkIv (password)

Reviewer #2 (Remarks on code availability)

I tried multiple times to access the demo instance using the “Credentials for demo instance” and unfortunately kept getting the same error message (see below). If there are alternate ways to access, I would be interested in testing it out and exploring the web app!

The server is temporarily unable to service your request due to maintenance downtime or capacity problems. Please try again later.

Response: Please see our response to the previous comment. Should you notice additional opportunities for improvement during your future use of the Knowledge Connector, we would appreciate your feedback for consideration in subsequent updates. Input from domain experts and the broader precision oncology community will be indispensable to the system's continuing refinement.